# Tunable Electronic Property and Robust Type-II Feature in Blue Phosphorene/MoSi$_2$N$_4$ Bilayer Heterostructure

**Xiaolin Cai** [1,*]**, Zhengwen Zhang** [1]**, Guoxing Chen** [1]**, Qin Wang** [1] **and Yu Jia** [2,3,*]

1 School of Physics and Electronic Information Engineering, Henan Polytechnic University, Jiaozuo 454000, China
2 Key Laboratory for Special Functional Materials of Ministry of Education, Collaborative Innovation Center of Nano Functional Materials and Applications, School of Material Science and Engineering, Henan University, Kaifeng 475004, China
3 International Laboratory for Quantum Functional Materials of Henan, School of Physics and Microelectronics, Zhengzhou University, Zhengzhou 450001, China
\* Correspondence: caixiaolin@hpu.edu.cn (X.C.); jiayu@henu.edu.cn (Y.J.)

**Abstract:** Constructing novel van der Waals heterostructures (vdWHs) is one of the effective methods for expanding the properties and applications of single materials. In this contribution, a blue phosphorene (Blue P)/MoSi$_2$N$_4$ vertical bilayer vdWH was constructed, and its crystal and electronic structures as well as optical properties were systematically studied via first principles calculation. It was found that the Blue P/MoSi$_2$N$_4$ vdWH with good thermal and dynamic stabilities belongs to the type-II indirect bandgap semiconductor with the bandgap of 1.92 eV, which can efficiently separate electrons and holes. Additionally, the two band edges straddle the redox potential of water, and the charge transfer follows the Z-scheme mode, making the Blue P/MoSi$_2$N$_4$ vdWH a promising catalyst of hydrogen production through splitting water. Meanwhile, the Blue P/MoSi$_2$N$_4$ vdWH has higher optical absorption than its two component monolayers. Both the external electric field and vertical strain can easily tailor the bandgap of Blue P/MoSi$_2$N$_4$ vdWH while still preserving its type-II heterostructure characteristics. Our proposed Blue P/MoSi$_2$N$_4$ vdWH is a promising photovoltaic two-dimensional material, and our findings provided theoretical support for the related experimental exploration.

**Keywords:** Blue P/MoSi$_2$N$_4$ vdWH; electronic properties; type-II; strain; *E*-field

## 1. Introduction

Since graphene was discovered, two-dimensional (2D) materials have become the focus of both theoretical and experimental investigations in light of their distinctive crystal structures and physical characteristics [1]. Various 2D materials, such as hexagonal boron nitride (h-BN) [2,3], black phosphorene [4–7] and transition-metal dichalcogenides (TMDCs) [8–10], which can be used as the versatile platforms for the applications of optoelectronics and electronics, have attracted extensive attention. [11–13].

Recently, the 2D van der Waals (vdW) layered material MoSi$_2$N$_4$ was experimentally prepared by Hong et al. [14] As a semiconductor, MoSi$_2$N$_4$ monolayer (ML) has an indirect bandgap of 2.41 eV, which can be achieved by inserting one MoN$_2$ ML into the middle of two SiN MLs, arranged in the form of a N–Si–N–Mo–N–Si–N seven-atomic layer. This unique crystal structure of MoSi$_2$N$_4$ ML determines its many novel features, such as extraordinary mechanical strength and stability, higher carrier mobility, and excellent electronic and valleytronic properties [15–18], granting it great potential for applications in photodetection, photovoltaics, and photocatalysis [19]. Additionally, 2D vertical MoSi$_2$N$_4$-based vdW heterostructures (vdWHs) can further enrich the properties of MoSi$_2$N$_4$ ML, broadening its applications [20–24]. Theoretical findings revealed that MoSi$_2$N$_4$ ML contacting with metal exhibits adjustable Schottky barrier height in a wind range and large

Schottky barrier height slope, superior to most other 2D semiconductors, which is owing to the two outlying Si-N sublayers protecting the semiconducting electronic states in the septuple-layer $MoSi_2N_4$ ML [21]. The $InSe/MoSi_2N_4$ vdWH shows the properties of a type-II band alignment and a favorable direct bandgap of 1.61 eV, high electron mobility ($10^4$ $cm^2V^{-1}s^{-1}$), obvious optical absorption ($10^5$ $cm^{-1}$) in the visible light range, and appropriate band edge values for overall water splitting [22]. Additionally, the $C_2N/MoSi_2N_4$ vdWH possesses ideal interface electronic properties, large interlayer charge transfer, and good visible light response and has great potential application in the field of photocatalytic water splitting [23]. The valleytronic properties of $MoSi_2N_4$ ML can been enhanced via forming the $MoSi_2N_4/CrCl_3$ bilayer vdWH [24]. Furthermore, the electronic properties of $MoSi_2N_4/GaN$ and $MoSi_2N_4/ZnO$ vdWHs can be substantially modified by applying an electric field (*E*-field) and strain, causing them to undergo transitions from type-I to type-II band alignment and from direct to indirect bandgap [20]. These findings demonstrate the potential applications of $MoSi_2N_4$-based vdWHs as an adjustable hybrid 2D material with enormous design flexibility in ultracompact optoelectronics.

Black phosphorene with an orthogonal structure has broad application prospects in nanoscale optoelectronic devices owing to the appropriate bandgap and excellent optical properties [25,26]. As an allotrope of black phosphorene, blue phosphorene (Blue P) has also aroused great research interest due to its unique optoelectronic properties [27,28]. Nonplanar Blue P ML, with a hexagonal crystal structure and semiconductor characteristics, has an indirect bandgap of 2 eV, possessing high carrier mobility ($10^3$ $cm^2/V \cdot s$) [29,30] and excellent optical absorption. Thus, ML Blue P is a commonly used 2D material for designing 2D vertical heterostructures with excellent properties through combination with other 2D materials [31,32]. The type-II Blue $P/MoSe_2$ vdWH has improved optical and electronic properties in contrast with the individual Blue P and $MoSe_2$ MLs [33]. The Blue $P/Mg(OH)_2$ heterobilayer can be used as a promising visible light photocatalyst for water splitting [34]. Through first principles calculations, Chen et al. found that the Blue $P/MoSi_2N_4$ vdWH is a potential photocatalytic candidate because of its direct Z-scheme characteristic and appropriate band edge values for overall water splitting [32]. However, they did not focus on the strain and *E*-field effects on the electronic property of Blue $P/MoSi_2N_4$ bilayer vdWH, which are vitally important for its application in semiconductor devices. Consequently, it is quite meaningful to investigate the response of its electronic property to *E*-field and strain and explore its other possibilities in future nanoelectronic devices.

In this contribution, using first principles calculations based on density functional theory (DFT), a 2D vertical Blue $P/MoSi_2N_4$ bilayer vdWH was designed, and its stacking configurations, stabilities, and electronic and optical properties were systematically investigated. The strain and *E*-field modified electronic properties of the Blue $P/MoSi_2N_4$ bilayer were also explored. It was shown that the Blue $P/MoSi_2N_4$ vdWH with good stability has the feature of type-II indirect bandgap vdWH with effectively separated electrons and holes, in which the two band edges straddle the redox potential of water, making it promising for hydrogen production through photocatalytic decomposition of water. Moreover, the Blue $P/MoSi_2N_4$ vdWH has enhanced optical absorption in contrast with the two individual MLs. Both the external *E*-field and vertical strain can easily tailor the bandgap of Blue $P/MoSi_2N_4$ vdWH while still preserving its type-II heterostructure characteristics. Our proposed Blue $P/MoSi_2N_4$ vdWH is a promising photovoltaic 2D material, and our findings offered theoretical support for the related experimental exploration.

## 2. Computational Details

All of our simulations were carried out on VASP [35] based on DFT developed by the hafner group at the University of Vienna in Austria, and the exchange–correlation interaction was described by the generalized gradient approximation (GGA) with the Perdew–Burke–Ernzerhof (PBE) functional proposed by Burke et al. in New Orleans of America [36]. A 500 eV energy cut-off was set for the plane wave expansion. The Grimme's DFT-D3 correction method [37,38] proposed by Grimme et al. in Münster

of Germany was adopted to estimate the long-range vdW interlayer interaction in the Blue P/$MoSi_2N_4$ vdWH. The convergence criteria for the total energy and the Hellmann–Feynman force on each atom were $10^{-5}$ eV and 0.01 eV/Å, respectively. The reciprocal space was sampled with the fine k-point meshes of $11 \times 11 \times 1$ and a $5 \times 5 \times 1$ in the Brillouin zone for the PBE calculations and the more expensive HSE06, respectively. The binding energy of the Blue P/$MoSi_2N_4$ vdWH was calculated based on the formula $E_b = E_{\text{Blue P}/MoSi_2N_4} - E_{\text{Blue P}} - E_{MoSi_2N_4}$, where $E_{\text{Blue P}/MoSi_2N_4}$, $E_{\text{Blue P}}$, and $E_{MoSi_2N_4}$ were the total energies of the Blue P/$MoSi_2N_4$ vdWH and isolated Blue P and $MoSi_2N_4$ MLs, respectively. In addition, the VASPKIT code [39] developed by Wang. in Xi'an of China was performed to process the calculated results of dielectric constant to obtain the optical absorption coefficient $A(\omega)$.

## 3. Results and Discussion

Firstly, the crystal and electronic structures of isolated Blue P and $MoSi_2N_4$ MLs were examined. Figure 1a,b showed the top and side views of the two isolated MLs, both hexagonal structures, and the optimized lattice constants are 3.27 Å and 2.90 Å for the Blue P and $MoSi_2N_4$ MLs (Table 1), respectively. The two corresponding electronic structures were shown in Figure S1a,b, respectively. The bandgaps estimated by the PBE (HSE) scheme are 1.95 eV (3.01 eV) for Blue P ML and 1.76 eV (2.41 eV) for $MoSi_2N_4$ ML, listed in Table 1, respectively. Both Blue P and $MoSi_2N_4$ MLs are indirect bandgap semiconductors. For Blue P ML, the conduction band minimum (CBM) is near the M point along the ΓM path, and the valence band maximum (VBM) is between the K and Γ points. $MoSi_2N_4$ ML has the CBM at the K point and the VBM at the Γ point. The above parameters are consistent with the previous studies [40–43].

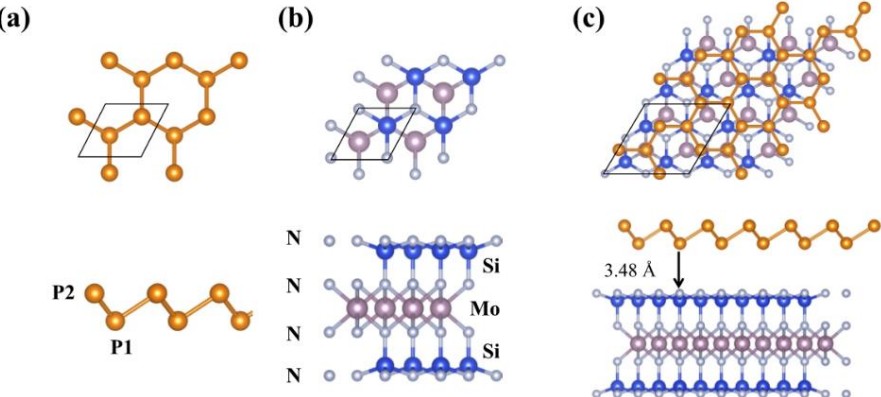

**Figure 1.** Top and side views of the crystal structures for (**a**) Blue P ML, (**b**) $MoSi_2N_4$ ML, and (**c**) Blue P/$MoSi_2N_4$ vdWH with the $T_1$ configuration. The rhombuses in the top views represent the unit cells of these 2D materials. The element symbols refer to the corresponding atomic layers in the Blue P and $MoSi_2N_4$ MLs in panels (**a,b**).

**Table 1.** Lattice constants, binding energies, interlayer distances, and bandgaps of the Blue P/$MoSi_2N_4$ bilayer with the four configurations together with the Blue P and $MoSi_2N_4$ MLs.

| | Blue P | $MoSi_2N_4$ | Blue P/$MoSi_2N_4$ | | | |
| | | | $T_1$ | $T_2$ | $B_1$ | $B_2$ |
|---|---|---|---|---|---|---|
| Lattice (Å) | 3.27 | 2.90 | 5.78 | 5.78 | 5.78 | 5.78 |
| $E_b$ (meV/Å$^2$) | / | / | −14.12 | −14.11 | −14.10 | −14.11 |
| $d_z$ (Å) | / | / | 3.46 | 3.45 | 3.48 | 3.47 |
| $E_g$ (eV) | 1.95/3.01 | 1.76/2.41 | 1.21/1.92 | 1.21 | 1.21 | 1.21 |

Based on the above calculations, the Blue P/$MoSi_2N_4$ bilayer vdWH was established through stacking the $\sqrt{3} \times \sqrt{3} \times 1$ supercell of Blue P ML on the $2 \times 2 \times 1$ supercell of $MoSi_2N_4$ ML. There is a lattice mismatch of 2.4% between the two component MLs.

The primitive cell of the Blue P/MoSi$_2$N$_4$ bilayer contains 34 atoms: 6 P, 16 N, 8 Si, and 4 Mo atoms, respectively. Four stacking configurations were considered: the two lower or upper P atoms in the Blue P ML located on the tops of the Mo and Si atoms or on the middle of the N and Si atoms in the MoSi$_2$N$_4$ ML—named T$_1$ and B$_1$ or T$_2$ and B$_2$ configurations, respectively—presented in Figures 1c and S3. After full relaxation, it was found that the four stacking configurations have the same lattice constants, 5.78 Å, and almost identical interlayer distances, 3.46, 3.45, 3.48, and 3.47 Å for T$_1$, T$_2$, B$_1$ and B$_2$ configurations, respectively. The binding energies of Blue P/MoSi$_2$N$_4$ vdWHs were calculated using the same method as those of the MoTe$_2$/PtS$_2$ [44] and MoS$_2$/Ga$_2$O$_3$ [45] vdWHs. The negative binding energies of the four configurations, which were obtained from the PBE+D3 functional, are all approximately −14 meV/Å$^2$, indicating their energy stability. Hybridization should also be included in the actual interlayer interactions of these vdWHs. If the hybridization were included in the calculations, the binding energy might be larger, very close to the vdW interaction ~−20 meV/Å$^2$. Thus, the vdW interaction dominates the interlayer interaction of Blue P/MoSi$_2$N$_4$ bilayer. More importantly, the four configurations have the same electronic properties, and all semiconductors have an indirect bandgap of 1.21 eV obtained from the PBE scheme. The parameters of the Blue P/MoSi$_2$N$_4$ vdWH were listed in Table 1. Given the above similarities of these four configurations, for simplicity, only the Blue P/MoSi$_2$N$_4$ vdWH with T$_1$ configuration was discussed in the following, shown in Figure 1c.

Next, the phonon spectrum and ab initio molecular dynamics (AIMD) simulations were studied to analyze the dynamic and thermal stabilities of Blue P/MoSi$_2$N$_4$ vdWH, as shown in Figure 2. No imaginary frequencies were observed in the phonon spectra (Figure 2a). The result for the AIMD simulations within 3 ps at 300 K illustrated that the structure of Blue P/MoSi$_2$N$_4$ bilayer is still preserved, and the average total energy per atom oscillates over a small range (Figure 2b). The above findings proved the dynamic and thermal stabilities of the Blue P/MoSi$_2$N$_4$ bilayer.

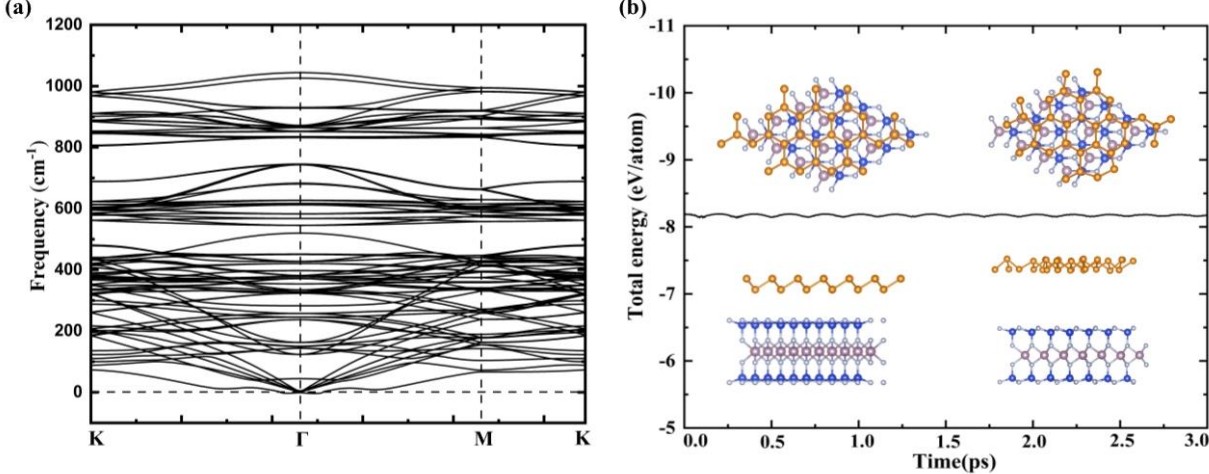

**Figure 2.** (**a**) Phonon spectra and (**b**) average total energy per atom vs time for the Blue P/MoSi$_2$N$_4$ vdWH. The left and right insets in panel (**b**) are the initial and final structures, respectively, of Blue P/MoSi$_2$N$_4$ vdWH during the AIMD simulations.

In order to facilitate the analysis of the electronic properties of Blue P/MoSi$_2$N$_4$ vdWH, the band structures were calculated for the $\sqrt{3} \times \sqrt{3} \times 1$ supercell of Blue P ML and the $2 \times 2 \times 1$ supercell of MoSi$_2$N$_4$ ML using the PBE functional presented in Figure 3a,b. It was found that the supercell has the same bandgap as the primitive cell (Figure S1), both 1.95 eV for Blue P ML and 1.76 eV for MoSi$_2$N$_4$ ML, respectively. Because of the band folding, the CBM and VBM transfer to near the K point and near the M point in the $\sqrt{3} \times \sqrt{3} \times 1$ supercell, respectively (Figure 3a). For the $2 \times 2 \times 1$ supercell of MoSi$_2$N$_4$ ML, the CBM and VBM are still located at the K and Γ points, respectively (Figure 3b),

because the K and M points fold to the K and Γ points, respectively. Furthermore, due to the heavy Mo atom in MoSi$_2$N$_4$ ML, the band structures with and without SOC were also plotted, which demonstrates that SOC only makes the band structure of MoSi$_2$N$_4$ ML split and does not change the bandgap and band edges (Figure S3). Figure 3c presented the projected band structure of the Blue P/MoSi$_2$N$_4$ vdWH. It was clearly revealed that the Blue P/MoSi$_2$N$_4$ vdWH is a semiconductor with the indirect bandgaps of 1.21 and 1.92 eV by the PBE and HSE schemes, shown in Figure 3a and Figure S4, respectively. Clearly, the component MoSi$_2$N$_4$ and Blue P MLs still basically keep their own original electronic features in the Blue P/MoSi$_2$N$_4$ vdWH. The CBM and VBM are located near the K point and at the Γ point, contributed by the Blue P and MoSi$_2$N$_4$ layers, respectively. This fully demonstrated that the Blue P/MoSi$_2$N$_4$ bilayer belongs to the typical type-II vdWH.

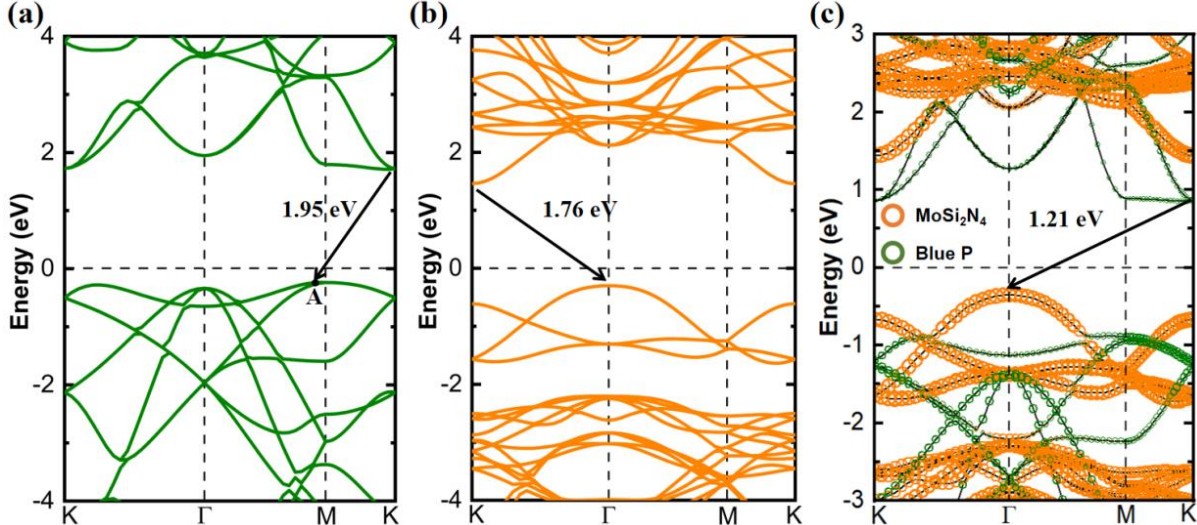

**Figure 3.** Band structures for (**a**) the $\sqrt{3} \times \sqrt{3} \times 1$ supercell of Blue P ML and (**b**) the $2 \times 2 \times 1$ supercell of MoSi$_2$N$_4$ ML. (**c**) Projected band structure of Blue P/MoSi$_2$N$_4$ vdWH. Here, the Fermi levels were set to zero.

The work function and the charge density difference were used to explain the formation mechanism of the 2D type-II Blue P/MoSi$_2$N$_4$ vdWH (see Figure S5). The calculated work functions are 5.93 eV for Blue P ML and 5.07 eV for MoSi$_2$N$_4$ ML, respectively, which are close to previous studies [42,46]. Thus, when the two 2D ML semiconductors contact to form the Blue P/MoSi$_2$N$_4$ vdWH, the electrons migrate from the MoSi$_2$N$_4$ ML to Blue P ML until they reach the same Fermi level, also affirmed by the differential charge density and the planar-averaged charge density difference along the z direction, demonstrated in Figure 4a,b, respectively. The charge density difference was calculated by the formula [47,48]: $\Delta\rho = \rho_{\text{Blue P/MoSi}_2\text{N}_4} - \rho_{\text{Blue P}} - \rho_{\text{MoSi}_2\text{N}_4}$, where $\rho_{\text{Blue P/MoSi}_2\text{N}_4}$, $\rho_{\text{Blue P}}$, and $\rho_{\text{MoSi}_2\text{N}_4}$ are the charge density distribution of Blue P/MoSi$_2$N$_4$ vdWH, Blue P, and MoSi$_2$N$_4$ MLs, respectively. The planar-averaged charge density difference along the z direction was obtained by integrating the above differential charge density in the 2D surface. Figure 4a manifested the charge gain and loss for Blue P (yellow) and MoSi$_2$N$_4$ (cyan) layers at the interface of Blue P/MoSi$_2$N$_4$ vdWH, leading to the positive and negative charges around the Blue P and MoSi$_2$N$_4$ MLs (Figure 4b), respectively. All the above results meant that electrons transfer from MoSi$_2$N$_4$ to Blue P ML in the interface of Blue P/MoSi$_2$N$_4$ vdWH, and the charge transfer amount is 0.0066 *e*, obtained from Bader charge analysis, signifying the weak vdW interaction between Blue P and MoSi$_2$N$_4$ MLs. Importantly, this charge transfer causes a built-in *E*-field from MoSi$_2$N$_4$ to Blue P ML across the interface of Blue P/MoSi$_2$N$_4$ vdWH.

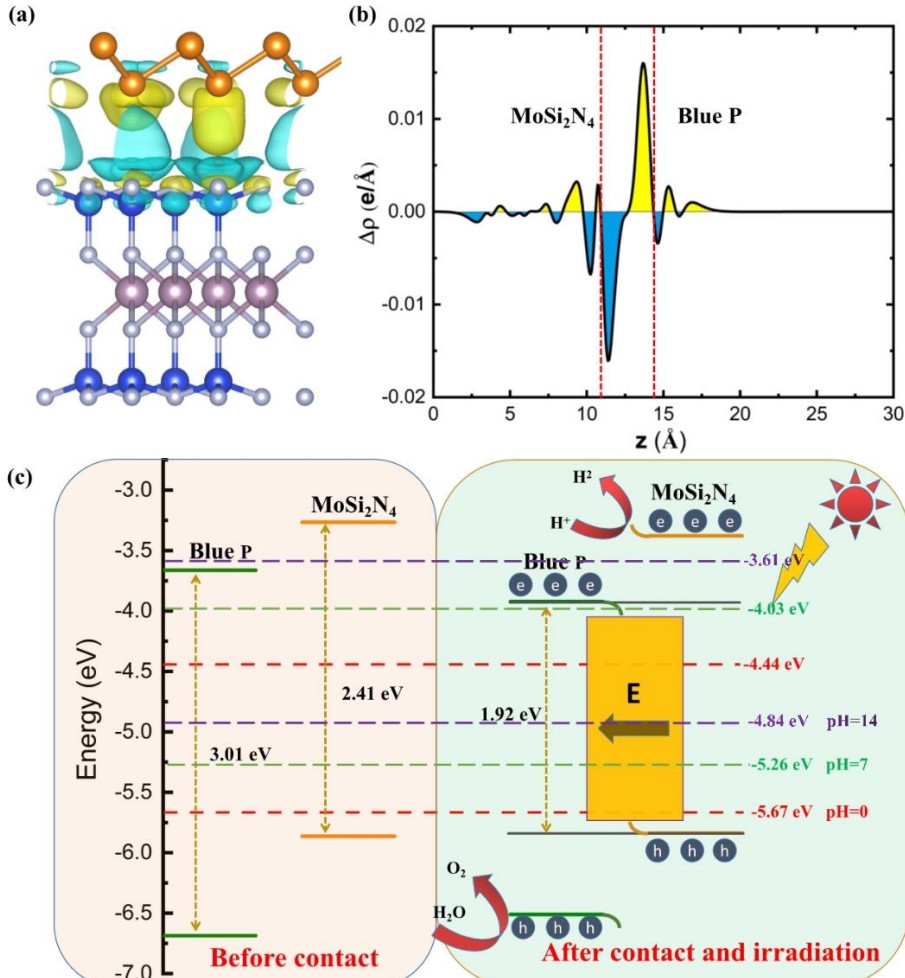

**Figure 4.** (**a**) Charge density difference (isosurface level: 0.0000745 $e/a_0^3$), (**b**) planar-averaged charge density difference along the z-axis, and (**c**) photocatalytic mechanism for the Blue P/MoSi$_2$N$_4$ vdWH.

Figure 4c showed the band alignments of MoSi$_2$N$_4$ and Blue P MLs as well as the redox potential of water with pH = 0, 7, and 14 relative to the vacuum level before and after contact. The reduction and oxidation potentials of water are affected by pH, which can be calculated by the following formulas:

$$E_{H^+/H_2O}^{red} = -4.44 \text{ eV} + pH \times 0.059 \text{ eV},$$
$$E_{O_2/H_2O}^{ox} = -5.67 \text{ eV} + pH \times 0.059 \text{ eV}.$$

Before the two MLs combine together, the CBM and VBM of MoSi$_2$N$_4$ ML are $-3.26$ and $-5.86$ eV, respectively, and the CBM and VBM of Blue P are $-3.66$ and $-6.69$ eV, respectively. By comparing the band edges and the redox potentials of water, it was found that the isolated Blue P and MoSi$_2$N$_4$ MLs can be used as catalysts for hydrogen production through splitting water from acidic to not-too-strong alkaline conditions and from acidic to alkaline conditions, respectively. However, as a result of the influence of the interlayer interaction and lattice mismatch, the bandgaps of the two component monolayers in the heterostructure are slightly changed, in contrast to those before contact [49,50]. In the Blue P/MoSi$_2$N$_4$ vdWH, the band edges of the MoSi$_2$N$_4$ ML are higher than those of the Blue P ML, and the CBM and VBM are $-3.37$ and $-6.51$ eV—mainly contributed by the Blue P and MoSi$_2$N$_4$ layers, respectively—which means that the Blue P/MoSi$_2$N$_4$ vdWH belongs to the type-II vdWH for hydrogen production through splitting water with pH = 0–7. Under the irradiation of incident light, the electrons transfer from the VBMs to the CBMs for the two component Blue P and MoSi$_2$N$_4$ layers, which is the typical charge transfer

mode of type-II heterostructures, such as 2D III-nitride/ZnO [51] and InX (X = S, Se)/YS$_2$ (Y = Mo, W) [52] vdWHs. Consequently, photogenerated holes and electrons accumulate in the VBMs and CBMs, respectively. Due to the band offsets, photogenerated electrons can migrate from the CBM of MoSi$_2$N$_4$ to the CBM of Blue P, while photogenerated holes follow a reverse migration. A built-in *E*-field from MoSi$_2$N$_4$ to blue P MLs is induced by this interlayer charge transfer, which can promote the recombination of the photogenerated electrons and holes in the CBM and the VBM of Blue P and MoSi$_2$N$_4$ MLs, respectively, while effectively separating the photogenerated carriers in the CBM and VBM of MoSi$_2$N$_4$ and Blue P MLs, respectively. Compared to the isolated Blue P and MoSi$_2$N$_4$ MLs, this Z-scheme and type-II Blue P/MoSi$_2$N$_4$ vdWH have significantly enhanced redox capacity because the hydrogen evolution reaction occurs in the CBM of MoSi$_2$N$_4$ layer with higher potential and the oxygen evolution reaction is carried out on the VBM of Blue P ML with lower potential.

The optical absorption property is critical for semiconductor applications in the photovoltaic field. The complex dielectric function was calculated to model the optical property, namely $\epsilon(\omega) = \epsilon_1(\omega) + i\epsilon(\omega)$, where $\omega$, $\epsilon_1(\omega)$ and $\epsilon_2(\omega)$ are the frequency of incident light and the real and imaginary parts of the dielectric function, respectively. The usual Kramers–Kronig transformation was performed to calculate the real part $\epsilon_1(\omega)$, and the imaginary part $\epsilon_2(\omega)$ was obtained through calculating the electronic ground state, a method which has been adopted by many theoretical studies [53,54]. For the 2D materials, the optical absorption is calculated using the formula $A(\omega) = \frac{\omega L \epsilon_2(\omega)}{c}$, where *c* and *L* are the speed of light in vacuum and the slab thickness of the simulated cell, respectively.

The optical absorption in the Blue P/MoSi$_2$N$_4$ vdWH varied with the wavelength from 200 to 800 nm in Figure 5, and those in the isolated Blue P and MoSi$_2$N$_4$ MLs were also given as a contrast. Interestingly, the Blue P/MoSi$_2$N$_4$ vdWH has significantly improved optical absorption relative to the isolated Blue P and MoSi$_2$N$_4$ MLs, particularly from 200 to 450 nm (ultraviolet to violet). Additionally, the absorption peak rose to 33.49% near 246 nm. This greatly enhanced optical absorption nature results from more atoms being involved in optical absorption and the decreased bandgap due to the interlayer coupling as well as the lattice mismatch of Blue P and MoSi$_2$N$_4$ MLs in the vdWH. The intriguing optical absorption can make the type-II Blue P/MoSi$_2$N$_4$ vdWH promising in photovoltaic devices [54].

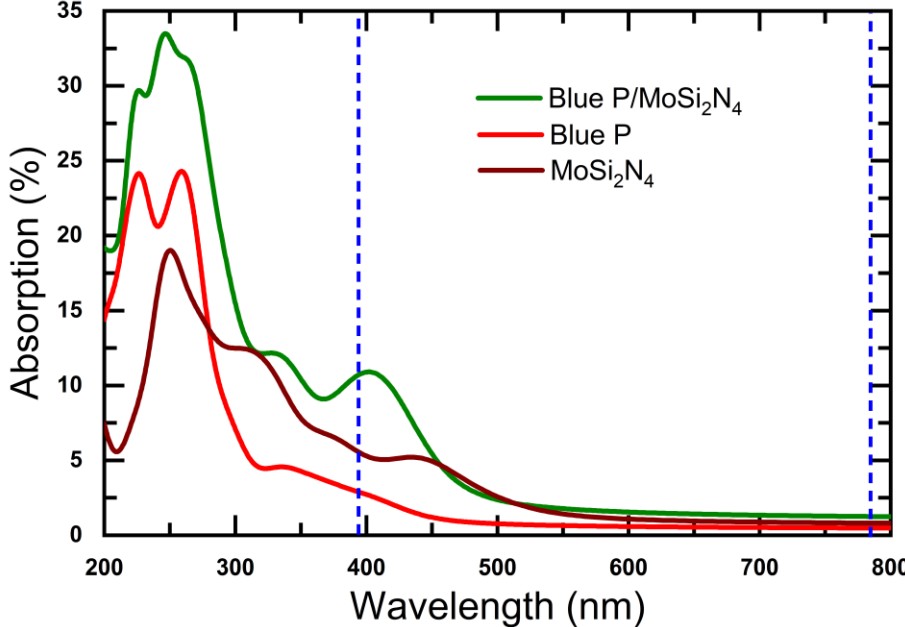

**Figure 5.** Optical absorption of the Blue P/MoSi$_2$N$_4$ vdWH together with those of isolated Blue P and MoSi$_2$N$_4$ MLs. The visible region is surrounded by blue dashed lines.

Having tunable electronic properties under *E*-field and strain is quite important for the practical applications of 2D materials. In this paper, the vertical *E*-field- and strain-induced electronic properties of Blue P/MoSi$_2$N$_4$ vdWH were explored. The positive direction of *E*-field was defined from MoSi$_2$N$_4$ to Blue P layer in the heterobilayer. Figure 6 presented the band alignment, bandgap, and band offset of Blue P/MoSi$_2$N$_4$ vdWH as a function of *E*-field. It was clearly revealed that the *E*-field can significantly regulate the band edge, bandgap, and band offset of Blue P/MoSi$_2$N$_4$ vdWH. The evolution of the CBM and VBM of Blue P and MoSi$_2$N$_4$ MLs in the vdWH as a function of *E*-field was shown in Figure 6a. Clearly, the CBM and VBM of Blue P ML are reduced almost linearly with the *E*-field increasing from −0.6 to 0.6 V/Å. The VBM of MoSi$_2$N$_4$ ML undergoes an increase with the *E*-field increasing from −0.6 to −0.45 V/Å, a sudden change with the *E*-field changing from −0.45 to −0.4 V/Å, and then a linear increase with the *E*-field from −0.45 to 0.6 V/Å, while the CBM of MoSi$_2$N$_4$ ML increased linearly all through the *E*-field from −0.6 to 0.6 V/Å. The physics behind the above variation of band edges with the *E*-field is the variations of band structures for the Blue P and MoSi$_2$N$_4$ MLs in the vdWH. The negative *E*-field can push electrons from MoSi$_2$N$_4$ to Blue P ML, causing the bands of MoSi$_2$N$_4$ and Blue P MLs to move downwards and upwards, shown in Figure 7a,b, respectively. Additionally, the positive *E*-field can produce exactly the opposite results, shown in Figure 7c,d. This *E*-field-induced band movement also changes the bandgap and the band offset in the Blue P/MoSi$_2$N$_4$ vdWH, shown in Figure 6b.

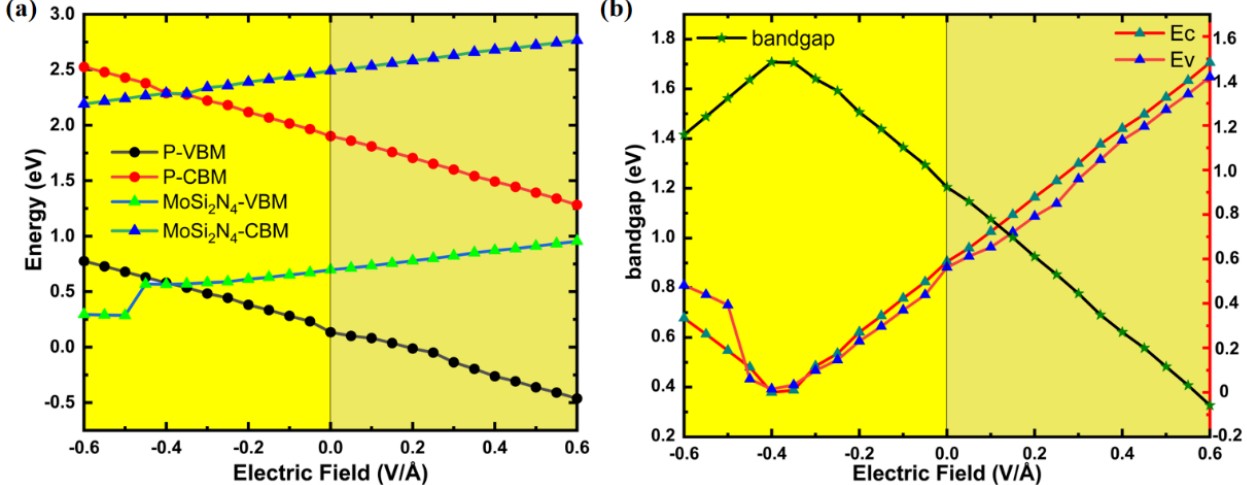

**Figure 6.** (**a**) Band alignment and (**b**) bandgap together with band offset as a function of *E*-field in the Blue P/MoSi$_2$N$_4$ vdWH.

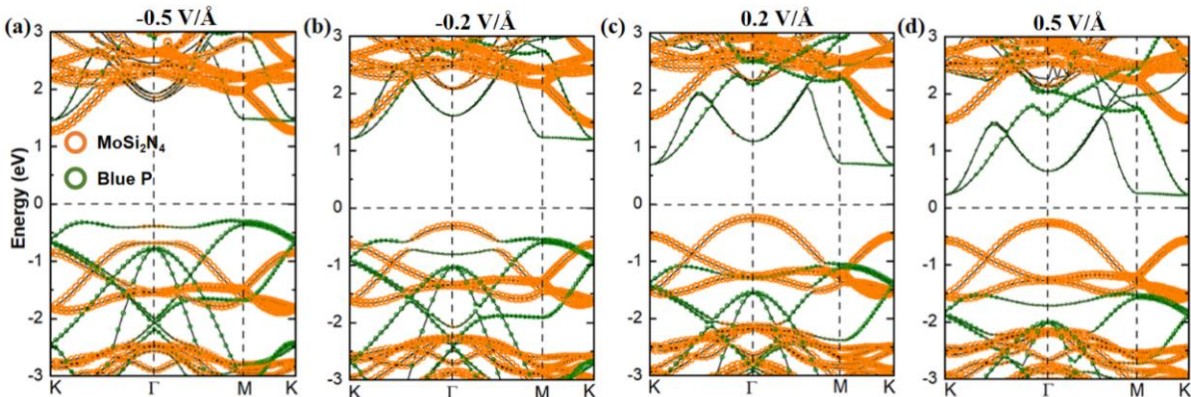

**Figure 7.** Projected band structures of the Blue P/MoSi$_2$N$_4$ vdWH under the *E*-fields of (**a**) −0.5, (**b**) −0.2, (**c**) 0.2, and (**d**) 0.5 V/Å. The Fermi levels were set to zero in these panels.

Without *E*-field, the CBM and VBM of Blue P/MoSi$_2$N$_4$ vdWH are from Blue P and MoSi$_2$N$_4$ MLs, respectively. Due to the CBM of Blue P ML and the VBM of MoSi$_2$N$_4$ ML moving face to face, the bandgap decreases from 1.21 to 0.32 eV as the positive *E*-field increases from 0.0 to 0.6 V/Å, accompanied by the increase in band offset. When the negative *E*-field is applied from 0–−0.35 V/Å, the bandgap monotonically increases from 1.21 to 1.71 eV since the CBM of Blue P and the VBM of MoSi$_2$N$_4$ shift backwards. Additionally, it retains 1.71 eV unchanged under the *E*-field from −0.35 to −0.40 V/Å, in which the two CBMs as well as the two VBMs meet each other in the two-component MLs. After this, the CBM and VBM of Blue P ML are both higher than those of MoSi$_2$N$_4$ ML, and the bandgap of Blue P/MoSi$_2$N$_4$ vdWH is determined by the CBM of MoSi$_2$N$_4$ ML and the VBM of Blue P ML. Under the *E*-field from −0.40 to −0.60 V/Å, the bandgap decreases from 1.71 to 1.41 eV, shown in Figure 6b. This band edge variation with the negative *E*-field causes the VBM and CBM band offsets to firstly decrease to zero and then increase, shown in Figure 6b.

Next, the evolution of the electronic property of Blue P/MoSi$_2$N$_4$ vdWH with the vertical strain was systemically investigated. The vertical strain was defined as $\varepsilon = \frac{d_z - d_{z_0}}{d_{z_0}} \times 100\%$, here $d_z$ and $d_{z_0}$ are the strained and unstrained interlayer distances between Blue P and MoSi$_2$N$_4$ MLs in the Blue P/MoSi$_2$N$_4$ vdWH, respectively. Namely, the vertical strain is positive tensile strain when the interlayer distance is larger than the unstrained one; otherwise, it is negative compressive strain. Here, the interlayer spacing between the two component MLs ranges from 2.86 to 4.46 Å with a step of 0.20 Å in the Blue P/MoSi$_2$N$_4$ vdWH, corresponding to the vertical strain from −19.4 to 28.9%. The evolution of electronic property for the Blue P/MoSi$_2$N$_4$ vdWH was shown in Figures 8 and 9. As the layer spacing decreases, the interaction between Blue P and MoSi$_2$N$_4$ MLs increases, and thus more charges are transferred from MoSi$_2$N$_4$ to the Blue P layer, causing their band structure to move downwards and upwards, presented in Figure 9a,b, respectively. Meanwhile, due to the induced strong interlayer interaction, the band structure of the two component MLs varies slightly compared to their isolated structures. The larger interlayer distance can decrease the interlayer interaction, reducing the charge transfer from MoSi$_2$N$_4$ to the Blue P layer, which makes the band structures of Blue P and MoSi$_2$N$_4$ MLs shift downwards and upwards, shown in Figure 9c,d, respectively. Therefore, it was clearly observed that when the interlayer spacing spans from 2.86 to 4.46 Å, the band edges of Blue P and MoSi$_2$N$_4$ MLs decrease and increase gradually in Figure 8a, respectively. Moreover, the bandgap of the Blue P/MoSi$_2$N$_4$ vdWH decreases from 1.36 to 1.07 eV while the band offsets are increased.

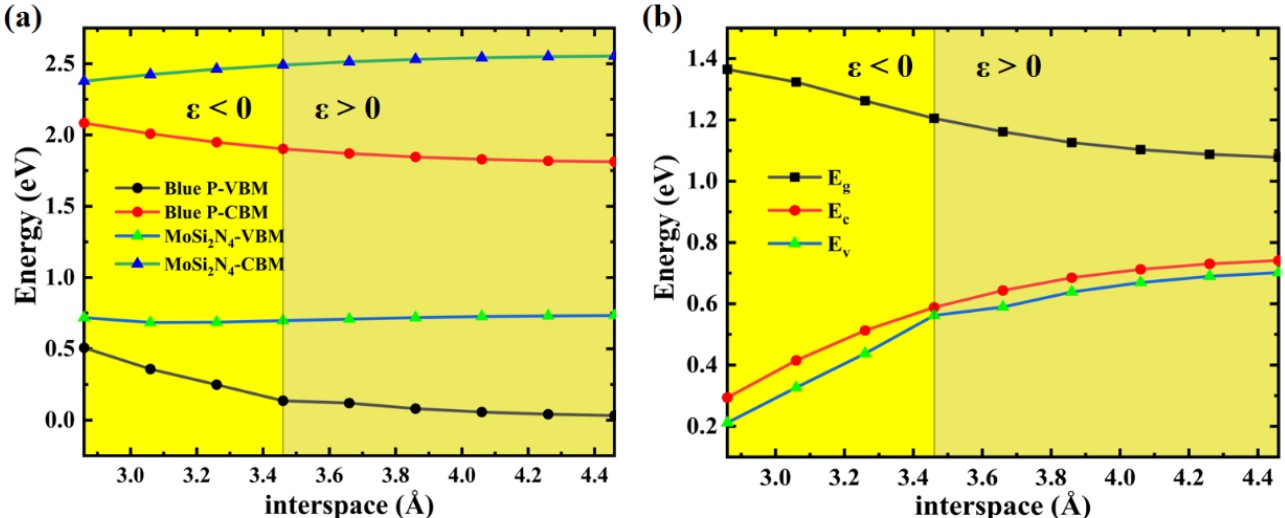

**Figure 8.** (**a**) Band alignment and (**b**) band offset together with bandgap as a function of interspace in the Blue P/MoSi$_2$N$_4$ vdWH.

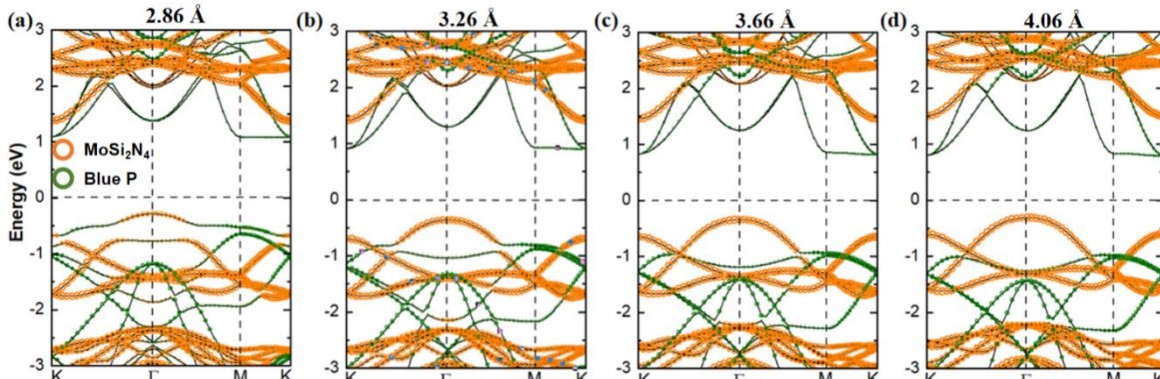

**Figure 9.** Projected band structures of the Blue P/MoSi$_2$N$_4$ vdWH under the vertical interspaces of (**a**) 2.86, (**b**) 3.26, (**c**) 3.66, and (**d**) 4.06 Å. In these panels, the Fermi levels were set to zero.

It was worth emphasizing that although the vertical *E*-field and strain can tailor the band structure of Blue P/MoSi$_2$N$_4$ vdWH, the type-II nature is robust, not tailored by *E*-field and strain. This manifested the broad application prospect of Blue P/MoSi$_2$N$_4$ vdWH in the photovoltaic field.

### 4. Summary

In conclusion, based on first principles calculations, a Blue P/MoSi$_2$N$_4$ vdWH was theoretically constructed, and its tunable electronic structure and optical properties were explored. It was found that the Blue P/MoSi$_2$N$_4$ vdWH is an indirect type-II vdWH with a bandgap of 1.92 eV and unique optical absorption, which can remarkably promote the separation of photogenerated carriers. Importantly, the band edges straddle the redox potentials of water, and the charge transfer is performed in a Z-scheme manner in the Blue P/MoSi$_2$N$_4$ vdWH, meaning that it is a potential photocatalyst with strong redox ability for water splitting. Both the vertical *E*-field and strain can easily tailor the bandgap of the Blue P/MoSi$_2$N$_4$ vdWH while still preserving its type-II heterostructure characteristics. Our proposed Blue P/MoSi$_2$N$_4$ vdWH is a promising photovoltaic 2D material, and our findings provided theoretical support for the related experimental exploration.

**Supplementary Materials:** The following supporting information can be downloaded at: https:// www.mdpi.com/article/10.3390/cryst12101407/s1, Figure S1: Band structures for the unit cells of (a) Blue P and (b) MoSi$_2$N$_4$ MLs. The black and red lines indicate the PBE and HSE results, respectively. Here, the Fermi levels are set to zero; Figure S2: Top and side views of the Blue P/MoSi$_2$N$_4$ vdWH with (a) T$_2$, (b) B$_1$, and (c) B$_2$ configurations; Figure S3: Electronic band structures for ML MoSi$_2$N$_4$ (a) without SOC and (b) with SOC. Here, the Fermi levels are set to zero, and the dark and red lines refer to the results obtained by the PBE and HSE schemes, respectively; Figure S4: Band structures for Blue P/MoSi$_2$N$_4$ vdWH by HSE scheme. Here, the Fermi level is set to zero.

**Author Contributions:** Conceptualization, X.C. and Y.J.; methodology, X.C.; software, Y.J.; validation, X.C., G.C. and Z.Z.; formal analysis, X.C. and Q.W; investigation, Z.Z.; resources, Y.J.; data curation, Z.Z., G.C. and Q.W.; writing—original draft preparation, X.C.; writing—review and editing, X.C.; visualization, Z.Z.; supervision, Y.J.; project administration, Y.J.; funding acquisition, Y.J. and X.C. All authors have read and agreed to the published version of the manuscript.

**Funding:** This research was funded by the National Natural Science Foundation of China, grant numbers 11804082, 12074102, and 11774078, and by the Doctoral Foundation of Henan Polytechnic University, grant number B2018-37.

**Data Availability Statement:** The data that supports the findings of this study are available from within the article and its Supplementary Materials.

**Conflicts of Interest:** The authors declare no conflict of interest.

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
