# Peer review of "Tunable Electronic Property and Robust Type-II Feature in Blue Phosphorene/MoSi2N4 Bilayer Heterostructure"

_crystals, doi:10.3390/cryst12101407_

Round 1

Reviewer 1 Report

Authors constructed a Blue P/MoSi2N4 vdWH and explored its tunable electronic structure based on first-principles calculations. The band edges straddle the redox potentials of water, and the charge transfer is performed in a Z-Scheme manner in the Blue P/MoSi2N4 vdWH.  The results are benefit for the experiment for the growth of Blue P/MoSi2N4 heterostructure. This job should be accepted in present form.

Author Response

We are very grateful to the reviewer for carefully reading and commenting our article, especially giving us these positive opinions.

Reviewer 2 Report

The authors should change the tense of the document and eliminate the use of "we".  The discussion should also include experimental results published by others in order to start the validation of the theoretical projections.

Author Response

Thank you for your constructive comments in our article. In the revised version, we have changed the tense of the article to the past formula, and changed the description of the objective laws to simple present tense, and also changed the statement containing “we” to a passive voice and eliminating “we”.

Currently, there are relatively few experimental studies on the 2D MoSi2N4 monolayer, and the reference we have found is its preparation by the chemical vapor deposition method in 2020 [Ref. 14, Science 369, 670–674 (2020)], and the experimental studies on the related van der Waals heterostructures has not been seen. However, we believe that for 2D monolayer MoSi2N4, as an excellent 2D material, there must be many excellent scientists performing experimental explorations. In the future, we can see a lot of relevant experimental results.

Reviewer 3 Report

In this paper, the authors reported a theoretical study of 2D van der Waals heterostructure based on MoSi2N4 and BlueP as a photocatalyst for water splitting. This van der Waals heterostructure has a type-II band structure and decent band edge positions for redox reaction. Furthermore, several intrinsic advantages, such as large specific surface area, tunable electronic performance and excellent optical adsorption, have been proved for the MoSi2N4/BlueP vdW heterostructure, which demonstrates the vdW heterostructure proposed by authors can be an efficient photocatalyst for water splitting. This work is significant and novel to provide a theoretical method to design 2D vdW heterostructures used as photocatalysts for water splitting. So the manuscript can be acceptable for publication of the Crystals after the minor revision as follows.

In Figure 1, the color of the P and Mo atoms should be distinguished.

How does the author determine that the MoSi2N and BlueP monolayers form by van der Waals forces? The comparation of the binding energy of the MoSi2N4/BlueP heterostructure with other reported heterostructures is needed such as Crystals 2022, 12, 425.

In practice, experiment is carried out in the non-zero pH solution. So the authors could include pH value impacts on the band alignments. In such a way, the prediction could be more reasonable.

More details of the optical calculations are missing, the dielectric function should be added, especially its real and imaginary parts, which are critical for understanding optical properties seeing Nanoscale, 2022,14, 8463-8473.

Author Response

We are very grateful to the reviewer for carefully reading and giving us these positive opinions.

Author Response

(The authors gave the same response as above.)

Round 2

Reviewer 3 Report

The manuscript can be accepted.

Reviewer 4 Report

It can be accepted now.